# Evidence that synthetic lethality underlies the mutual exclusivity of oncogenic KRAS and EGFR mutations in lung adenocarcinoma

Arun M Unni*[†], William W Lockwood*[†‡], Kreshnik Zejnullahu, Shih-Queen Lee-Lin, Harold Varmus*[§]

Cancer Biology Section, Cancer Genetics Branch, National Human Genome Research Institute, Bethesda, United States

*For correspondence: arun. unni@nih.gov (AMU); wlockwood@bccrc.ca (WWL); varmus@med.cornell.edu (HV)

[†]These authors contributed equally to this work

**Present address:** [‡]Department of Integrative Oncology, British Columbia Cancer Research Centre, Vancouver, Canada; [§]Meyer Cancer Center, Weill Cornell Medical College, New York, United States

**Competing interests:** The authors declare that no competing interests exist.

**Reviewing editor**: Chi Van Dang, University of Pennsylvania, United States

**Abstract** Human lung adenocarcinomas (LUAD) contain mutations in *EGFR* in ~15% of cases and in *KRAS* in ~30%, yet no individual adenocarcinoma appears to carry activating mutations in both genes, a finding we have confirmed by re-analysis of data from over 600 LUAD. Here we provide evidence that co-occurrence of mutations in these two genes is deleterious. In transgenic mice programmed to express both mutant oncogenes in the lung epithelium, the resulting tumors express only one oncogene. We also show that forced expression of a second oncogene in human cancer cell lines with an endogenous mutated oncogene is deleterious. The most prominent features accompanying loss of cell viability were vacuolization, other changes in cell morphology, and increased macropinocytosis. Activation of ERK, p38 and JNK in the dying cells suggests that an overly active MAPK signaling pathway may mediate the phenotype. Together, our findings indicate that mutual exclusivity of oncogenic mutations may reveal unexpected vulnerabilities and therapeutic possibilities.

## Introduction

Large-scale sequencing of cancer genomes has provided a unique opportunity to survey and interpret the genotype of common and rare tumors. These efforts have revealed mutations in well-known tumor suppressor genes and proto-oncogenes; in genes with normal functions not previously associated with neoplasia (such as RNA splicing and chromatin modification); and in genes unlikely to have any role in carcinogenesis (putative 'passenger mutations') (*Kandoth et al., 2013*; *Hoadley et al., 2014*). In several tumor types, genomic studies have revealed alterations in specific genes or signaling pathways that are highly associated with tumor origins, such as mutations affecting HIF-1 signaling in renal clear cell carcinoma (*Cancer Genome Atlas Research Network, 2013*), in the Wnt signaling pathway in colorectal carcinoma (*Cancer Genome Atlas Network, 2012*), and, more broadly, in the growth factor receptor-RAS-PIK3CA or–AKT pathways in a variety of cancers including lung adenocarcinoma (*Kandoth et al., 2013*; *Cancer Genome Atlas Research Network, 2014*). These studies have been vital for understanding the genetic mechanisms driving tumorigenesis and revealing new targets for therapeutic intervention.

However, these initial analyses are just beginning to explore more complex issues such as the co-incidences and temporal sequences of mutations, which may reveal processes driving tumor evolution and influence new strategies for targeted therapy (*Wong et al., 2014*). For example, numerous investigators have noted the apparent 'mutual exclusivity' of oncogenic alleles of

**eLife digest** A person develops cancer when changes in a cell's DNA (called mutations) allow the cell to grow rapidly and spread around the body. The mutated genes are often involved in controlling the growth of cells, such as two genes called *EGFR* and *KRAS,* which are associated with forms of lung cancer.

In a type of lung cancer called adenocarcinoma, the *KRAS* gene is mutated in about one-third of tumors and the *EGFR* gene is mutated in about 15%. However, the two mutations rarely or never occur in the same tumor. This could be because the effects of the mutations overlap, so that cells with both mutations have no advantages over cells with just one. Alternatively, it is possible that having both mutations may be harmful to tumor cells.

Here, Unni, Lockwood et al. analyzed genetic data from over 600 lung tumors and confirmed that none of them have cancer-causing mutations in both KRAS and EGFR. Then, Unni, Lockwood et al. carried out experiments using genetically engineered mice with mutated forms of both *KRAS* and *EGFR* that are activated by a drug called doxycycline. As expected, the mice developed lung tumors when exposed to the drug, but these tumors didn't grow any faster than mouse tumors that had mutations in only one of the genes. In the mice with both mutant genes, only one of the two genes was actually active in most of the tumor cells.

Unni, Lockwood et al. manipulated human lung tumor cells in the laboratory so that the cells had mutated versions of both genes. These cells developed serious abnormalities and died, which may be due to the over-activation of a communication pathway within the cells called MAPK signaling. The next challenges are to understand why the combination of these two mutant genes kills these cancer cells and to look for other combinations of mutations that can be toxic to cancer cells. In the future, it might be possible to develop drugs that can mimic the effects of these gene mutations to treat cancers.

well-known proto-oncogenes in certain types of cancer, but, aside from a few instances (*Petti et al., 2006*; *Sensi et al., 2006*), without experimentally verified explanations.

One of the first and most apparent of these mutually exclusive mutational combinations involves two well-studied proto-oncogenes, *KRAS* and *EGFR*, in human lung adenocarcinomas (LUAD). This finding is especially provocative because of the frequencies of mutant alleles of *KRAS* and *EGFR* occurring separately in LUAD: ~30% for *KRAS* mutations and ~15% for *EGFR* mutations (*Cancer Genome Atlas Research Network, 2014*). The explanation generally provided for the mutual exclusivity is that the products of these two loci operate in the same or overlapping signaling pathways and hence are functionally redundant. However, this idea has not been experimentally tested, and there is reason to question its validity. For example, since they are differently positioned in a signaling network (EGFR senses and transmits external signals from the cell membrane; RAS serves as a cytosolic node), the consequences of the mutations would not be expected to be identical. Hence mutations in both genes might be expected to offer a selective advantage over a mutation in just one of them. Furthermore, and tellingly, KRAS mutations are rarely, if ever, encountered in mutant EGFR-driven tumors that become resistant to treatment with tyrosine kinase inhibitors (TKIs), when a 'downstream' mutation in *KRAS* would be expected to provide a strong selective advantage (*Yu et al., 2013*).

One way to reconcile these observations is to propose that oncogenic mutation of both of these genes in the same cell confers an unexpected inhibitory phenotype, such as synthetic lethality, that is selected against during tumor development. In this study, we aimed to determine whether an inhibitory phenotype can account for the mutual exclusivity of *KRAS* and *EGFR* mutations in LUAD and to characterize the specific cellular effects that result from co-expression of both mutant alleles in lung cancer cells. Through analysis of available sequences of human tumor samples, generation of transgenic mouse models that express mutant KRAS and EGFR in the lung epithelium, and functional tests of these mutant genes in cultured tumor cells, we conclude that synthetic lethality is responsible for the mutually exclusive nature of activating mutations in these genes. We propose that this type of gene interaction may reveal vulnerabilities in lung and other cancers that suggest novel therapeutic strategies.

## Results

### EGFR and KRAS mutations are mutually exclusive in lung adenocarcinoma

Although oncogenic mutations in *KRAS* and *EGFR* occur in a significant fraction of human LUAD, they are rarely—if ever—observed together in the same tumor. Since these two genes are in the same pathway and activate similar downstream targets, it is generally assumed that there is no selective pressure to favor cells with both mutations over cells with a mutation in one of them. If complete functional redundancy explains the mutual exclusivity of these mutations, assessing a large panel of samples might reveal tumors with both genes mutated. To this end, we reviewed published somatic mutation data for *EGFR* and *KRAS* from 662 LUAD that were profiled as part of four separate studies (*Ding et al., 2008*; *Imielinski et al., 2012*; *Seo et al., 2012*; *Cancer Genome Atlas Research Network, 2014*). We excluded data from studies that profiled only specific exons or mutation hotspots in these two genes, as restricted sequencing might not detect recently characterized oncogenic mutations such as those in exon 20 of *EGFR* and codon 61 of *KRAS* or changes in codons not associated with gene activation. In total, 186 (28.2%) and 110 (16.6%) tumors had somatic mutations that changed the amino acid sequences (non-silent mutations) of KRAS and EGFR proteins, respectively—frequencies similar to those previously reported (*Heist and Engelman, 2012*). Moreover, all mutations in *KRAS* and 93.6% of *EGFR* mutations were identified as oncogenic (see 'Materials and methods', *Supplementary file 1*).

Attempts to estimate the expected frequency of finding coincident mutations in any two genes in a tumor cell population are influenced by a variety of potentially confounding factors, such as the size and transcriptional activity of the genes; mutation rates for cells in the tumor lineage and for individual genes; the point in the evolution of a tumor when a mutation occurs (since that will affect the percentage of cells containing the mutant allele); and the selective advantage or disadvantage of the mutation for the growth and survival of the cell in which it occurred (*Lawrence et al., 2013*, *2014*). These factors make predictions complex. In the simplest terms, if we define mutations in these genes as independent events with similar selective advantage that occur at a specific frequency across the population of tumors—and assume that they are not influenced by the factors stated above—the expected co-incidence of the mutations can be derived from a simple calculation (used in *Figure 1*, panel A). On these terms, we would anticipate that that approximately 31 tumors in the cohort presented in *Figure 1* would contain mutations in both genes. However, we observed only a single tumor in which this was the case; moreover, the EGFR mutation (R574L) in this tumor (which also harbored a KRAS$^{G12V}$ mutation) has not been previously identified as oncogenic, lies outside the kinase domain and hence is unlikely to be activating (*Figure 1A*). A left-tailed 2X2 Fisher's Exact Test indicates a significant negative association between the mutations (p = 8.62e$^{-17}$), confirming the mutually exclusive nature of these mutant alleles. Furthermore, even when analyzing tumors from smokers and never-smokers separately—which are known to have different frequencies of mutations in *KRAS* and *EGFR*—a statistically significant negative association between the co-occurrence of mutations in these genes was still observed (p = 4.43e$^{-10}$ and p = 0.002, respectively) (*Figure 1*). Our failure to identify any tumors with co-existing oncogenic mutations of these two genes in this combined data set strengthened the possibility that the observed mutual exclusivity might imply more than functional redundancy—for example, a toxic effect of co-existing oncogenic mutations.

### Mutual exclusive relationship between EGFR and KRAS mutations is unique in lung adenocarcinoma

Because the co-existence of mutations may be affected by numerous factors (e.g., mutation rates, sensitivity of detection, etc, as detailed above) and because relevant data sets are relatively small, we looked more broadly for combinations that might provide a selective advantage or disadvantage, including mutual exclusivities, in LUAD. To do this, we devised a method that explores whole-exome DNA sequencing data on a global scale. Tumors were stratified based on the somatic mutation status of a known proto-oncogene (e.g., *KRAS* or *EGFR*, as well as other genes). The number of tumors containing mutations in both the specified proto-oncogene and every other gene in the genome was then calculated on a gene-by-gene basis (*Figure 1B–H*).

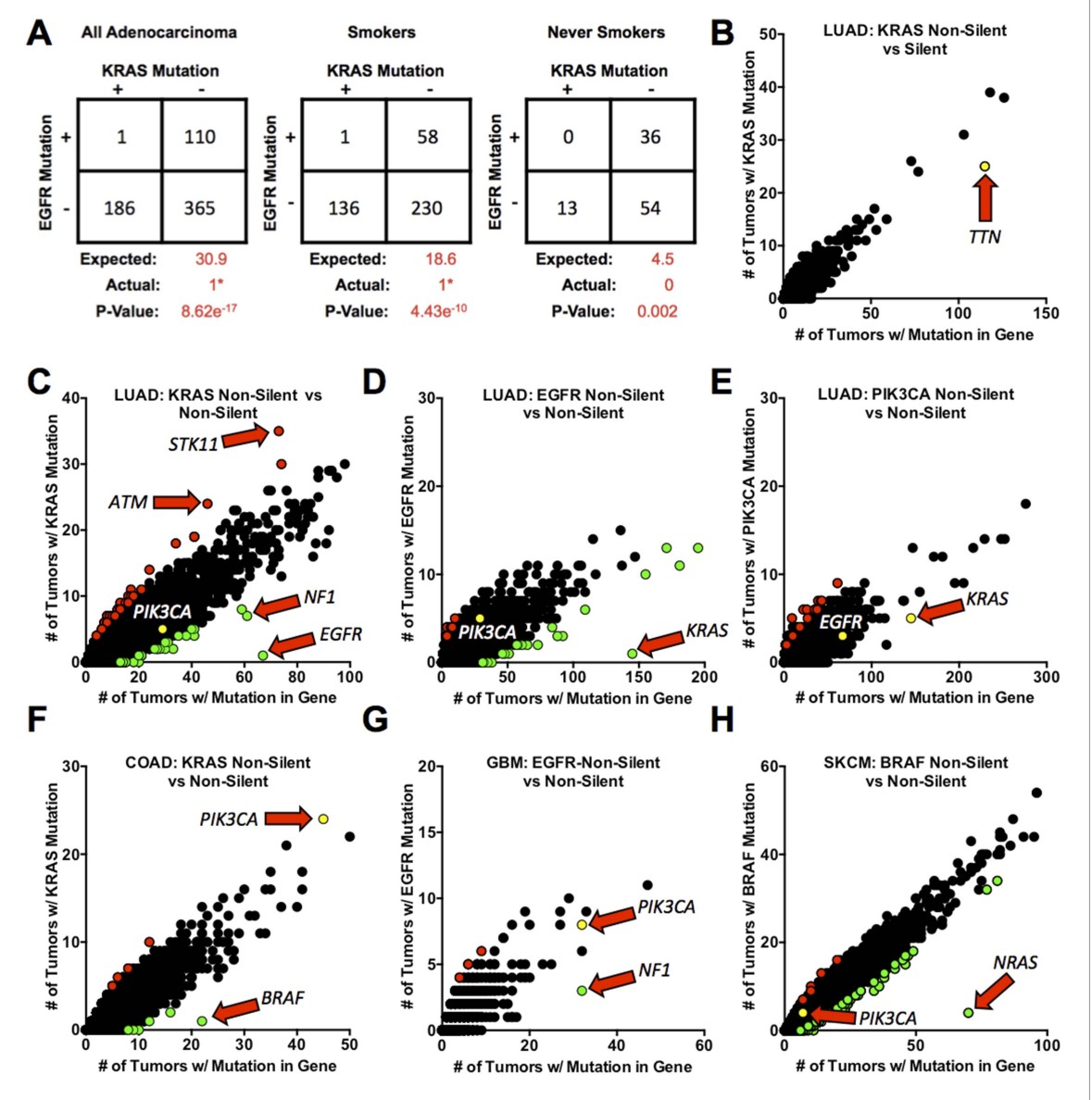

**Figure 1**. Mutations in KRAS and EGFR in lung adenocarcinoma are mutually exclusive. (**A**) Coincidence of *EGFR* and *KRAS* mutations in 662 lung adenocarcinomas (LUAD). Numbers are presented for all tumors combined and for tumors found in smokers or never smokers. 'Expected' values are for co-occurrence of mutations based on the frequency of mutation of either gene alone (including assumptions about selective advantage and other features as described in the text). (e.g., in all LUAD, *KRAS* is mutant in 28.2% and *EGFR* is mutant in 16.6%, so the expected number of co-occurrences is 30.9 cases or 0.282 × 0.166 × 662.). p-values were calculated by a left-tailed 2 × 2 Fisher's Exact Test to assess significance of negative association. *The single tumor with co-incident non-silent mutations contained a non-activating *EGFR*$^{R574L}$ mutation. (**B–H**) Frequencies of coincidence of somatic silent (synonymous point mutations) or non-silent mutations (non-synonymous point mutations and small insertions or deletions that affect protein coding sequence) in coding sequences of other genes co-occurring with non-silent mutations in proto-oncogenes (*EGFR, KRAS, PIK3CA,* or *BRAF*) in 520 LUAD or other cancer types as noted below for individual panels. In these panels, each point represents a unique gene but data points overlap when multiple genes are mutated in the same number of tumors. Red dots represent genes for which non-silent mutations are positively associated with proto-oncogene mutations in the tumors (Right side Fisher's Exact Test p ≤ 0.01); green dots represent genes for which mutations are negatively associated with mutant proto-oncogenes (Left side Fisher's Exact Test p ≤ 0.01). Yellow dots represent some important genes that are co-mutated at the expected rate. Genes of special interest are indicated by the arrows. (**B**) Co-occurrence of silent mutations in other genes with non-silent *KRAS* mutations in LUAD. (**C**) The similar analysis for non-silent mutations co-occurring with *KRAS* mutations. (**D, E**) As in panel **C**, but with mutant *EGFR* and *PIK3CA* in LUAD. (**F–H**) Additional analyses for mutations co-existing with mutant *KRAS* in colorectal adenocarcinoma (COAD), **F**; with mutant *EGFR* in glioblastoma multiforme (GBM), **G**; and with mutant *BRAF* in skin cutaneous melanoma (SKCM), **H**. For presentation purposes, not all gene points are displayed. Complete lists of significant gene pairs from each of analysis can be found in *Supplementary file 2*.

In the absence of selective pressures resulting from a second mutation, it would be anticipated that the more times a gene is mutated in the sample set, the more often it should co-occur in a tumor with a mutation in the proto-oncogene. Plotting these values on the y-axis against the total number of tumors with a mutation in those second genes on the x-axis would be expected to reveal a linear relationship. However, if selective pressures are involved, so that dual mutation enhances or restricts cell growth or survival, the value for a given gene will fall either above or below this line. If the additive effect confers a selective advantage, and a mutation of the second gene occurs more frequently in combination with the mutant proto-oncogene than expected, the result is displayed as a red dot (*Figure 1*; see legend for details). If a mutation in the second gene occurs less frequently than expected in combination with the mutant proto-oncogene, it is presumed to provide a selective disadvantage and the result is shown as a green dot in *Figure 1*.

From these so-called 'exclusivity plots', left and right 2 × 2 Fisher's Exact Tests can be performed between mutations in each gene and the proto-oncogene to confirm the degree of enhanced or diminished co-occurrence and, correspondingly, the potential degree of positive and negative selection conferred by the combinations. However, the confidence in any selective effect, reflected by a low p-value, should not be confused with the strength of the effect, since the number of mutations will influence the p-value. Therefore, since the numbers of mutations for the great majority of genes are relatively small in our data sets, we can place only a modest degree of confidence on the selective effects. However, in some cases, for example, TP53 plus KRAS mutations, there may be a highly significant but still relatively minor deviation from expected rates of co-occurrence. The deviations of greatest interest are therefore caused by those combinations of mutations, such as KRAS plus EGFR, that produce a major deviation from expectation at a high level of confidence because both genes are frequently mutant (*Figure 1* and *Supplementary file 2*; discussed further in 'Materials and methods').

To test the validity of this approach, we first assessed the frequency of synonymous mutations in many genes and their co-occurrence with non-silent mutations in KRAS—non-synonymous point mutations and small insertions and deletions—in LUAD. (Details of the analysis are provided in 'Materials and methods'.) Synonymous mutations—point mutations that do not change the amino acid sequence of a protein, also referred to here as 'silent' mutations—should be neutral (i.e., offer no selective advantage or disadvantage). Thus the more frequently a given gene contains a silent mutation, the more frequently such mutations should co-occur with a KRAS mutation, so that the number of co-mutations should increase in a linear fashion as described above.

To test these possibilities, we used whole-exome sequence data from an expanded set of 520 LUAD profiled as part of The Cancer Genome Atlas (TCGA) initiative (including 229 that were used in *Figure 1A*). Of these tumors, 145 had a somatic, non-silent mutation in KRAS. Silent mutations in any other gene co-occurred with a KRAS mutation as expected based on their respective frequencies (*Figure 1B*). For example, TTN, one of the largest genes in the genome, is known to have a very high mutation rate in lung cancer (*Lawrence et al., 2014*). This gene incurred frequent silent mutations in lung adenocarcinoma (indicated by a yellow dot) that overlapped with mutations in KRAS, much as anticipated (*Figure 1B*).

When this analysis was repeated, however, with non-silent mutations in genes, numerous statistically significant outliers were uncovered. For these genes, the number of mutations co-occurred more frequently or less than expected in concert with non-silent mutations in KRAS (*Figure 1C*, *Supplementary file 2*). EGFR is the gene with a co-mutation rate most significantly different from the expected number, demonstrating a negative association with mutant KRAS. Likewise, when this analysis was reversed by stratifying samples based on non-silent EGFR mutations (n = 67), KRAS is the only major outlier, demonstrating a similar negative association (*Figure 1C*).

The apparently negative association of KRAS and EGFR mutations is not a general phenomenon among known lung cancer proto-oncogenes. For example, PIK3CA mutations co-occurred as frequently as anticipated with either KRAS or EGFR mutations (as indicated by the yellow dot in *Figure 1C,D*). Likewise, non-silent mutations in other genes, including known oncogenes, were never observed to have a negative association with PIK3CA mutations in the same sample set, although several such mutations had a slight positive association (*Figure 1E*). Numerous genes that showed a similar frequency of non-silent mutations as EGFR and KRAS among the 520 samples co-occurred with mutations in KRAS or EGFR, within the range expected by chance in the absence of selective pressures (*Figure 1C,D*). This indicates that potential confounding factors, such as mutation rates and sensitivities of sequencing methods, are unlikely to be responsible for the data implying the exclusivity of mutations in KRAS and

*EGFR*; a more general skewing of the data would be anticipated if this was the case. Furthermore, although one tumor contained mutations in both *KRAS* and *EGFR*, the *KRAS* mutation, D33E, is not known to be activating. (Note that the tumor with KRAS$^{G12V}$ and EGFR$^{R574L}$ in *Figure 1A* was not part of this expanded dataset.) This suggests that mutations in both genes can occur and be detected by sequencing, but that activated alleles of both genes very rarely or never co-exist.

Our analysis also revealed that mutations in a few genes—such as *STK11* (also known as *LKB1*) and *ATM*, genes known to co-operate with mutant *KRAS* during lung tumorigenesis (*Ji et al., 2007*; *Efeyan et al., 2009*)—are positively associated with *KRAS* mutations (*Figure 1C*). The identification of genes known to provide a selective advantage to cancer cells driven by mutant *KRAS* provides strong evidence of the ability of this approach to uncover gene combinations under selective pressure during tumor evolution. Together, the analysis in *Figure 1* revealed that no other mutated genes in lung adenocarcinoma demonstrate such a strong mutually exclusive relationship, as do mutant *KRAS* and *EGFR*, further suggesting negative selection against the combination during lung tumorigenesis.

## Oncogenic mutations in members of the EGFR-RAS-RAF signaling pathway are negatively associated in carcinomas from different tissues of origin

To determine whether mutual exclusivity between oncogenic alleles is limited to *KRAS* and *EGFR* in lung adenocarcinoma or whether it is a more general phenomenon affecting other cancer lineages, we examined sequencing data from other cancer types with the methodology employed in *Figure 1*, seeking pairs of mutant proto-oncogenes that are disfavored or incompatible with tumorigenesis. First, we focused on cancer types known to have frequent activating mutations in *KRAS* or *EGFR*: colorectal adenocarcinoma (COAD) and glioblastoma multiforme (GBM), respectively (*Cancer Genome Atlas Research Network, 2008*; *Cancer Genome Atlas Network, 2012*). 'Exclusivity plots' for COAD revealed mutations in *BRAF* to be negatively associated with mutations in *KRAS*, to a degree similar to that we observed for co-existing mutations of *EGFR* and *KRAS* in lung adenocarcinoma (*Figure 1F*). Likewise, mutations in *NF1*, a suppressor of RAS activity, were negatively associated with *EGFR* mutations in GBM, a tumor type in which activating deletions in the extracellular domain of EGFR are typically found (*Figure 1G*). Looking beyond combinations involving mutant *EGFR* or *KRAS*, we found that *BRAF* mutations in skin cutaneous melanoma (SKCM) were negatively associated with *NRAS* (*Figure 1H*). These findings suggest that certain tumor cells might not tolerate mutations in combinations of certain genes whose products operate within the EGFR-RAS-RAF signaling pathway.

## Co-induction of mutant Kras and EGFR transgenes in the mouse lung epithelium does not alter the rate of tumor development but the tumors express only a single oncogene

The analysis presented in *Figure 1* supports the idea that certain pairs of mutually exclusive mutations may be selected against during tumorigenesis in some cell lineages, especially the combination of *EGFR* and *KRAS* oncogenic mutations in LUAD. To test this hypothesis experimentally, we took advantage of genetically engineered mice that develop LUAD when TetO-regulated transgenes encoding murine *Kras-G12D* (*Kras$^{G12D}$*) or human *EGFR-Deletion Exon 19* (*EGFR$^{DEL}$*) are induced by doxycycline, which activates the rtTA transcriptional regulator expressed in type II airway epithelial cells through a Clara Cell Secretory Protein promoter (*CCSP*) (*Fisher et al., 2001*; *Politi et al., 2006*). We attempted to express both transgenic oncogenes in the same cells by feeding doxycycline to tri-transgenic mice (*CCSP-rtTA;TetO-Kras$^{G12D}$;TetO-EGFR$^{DEL}$*, referred to as *CCSP;Kras$^{G12D}$:EGFR$^{DEL}$*) that encode both of the oncogenic proteins and rtTA. If co-expression of mutant *Kras* and mutant *EGFR* confers a selective advantage, tumors should appear earlier and progress more quickly than in bi-transgenic animals (*CCSP;Kras$^{G12D}$* or *CCSP;EGFR$^{DEL}$*) harboring only one oncogenic transgene. If the combination has no advantage or disadvantage because of functional identity of mutant KRAS and EGFR, tumorigenesis should be no different than observed in lines that contain only one of the oncogenes. If the combination is deleterious, tumors should be prevented or grow slowly.

We found that LUAD appeared in tri-transgenic mice after doxycycline-induction with the properties (time of detection, rate of progression, and histological appearance) observed in bi-transgenic *CCSP;Kras$^{G12D}$* mice (*Figure 2A*, see *Supplementary file 2* for genotypes of all animals). This result would appear to support the hypothesis that the two mutant oncogenes are functionally

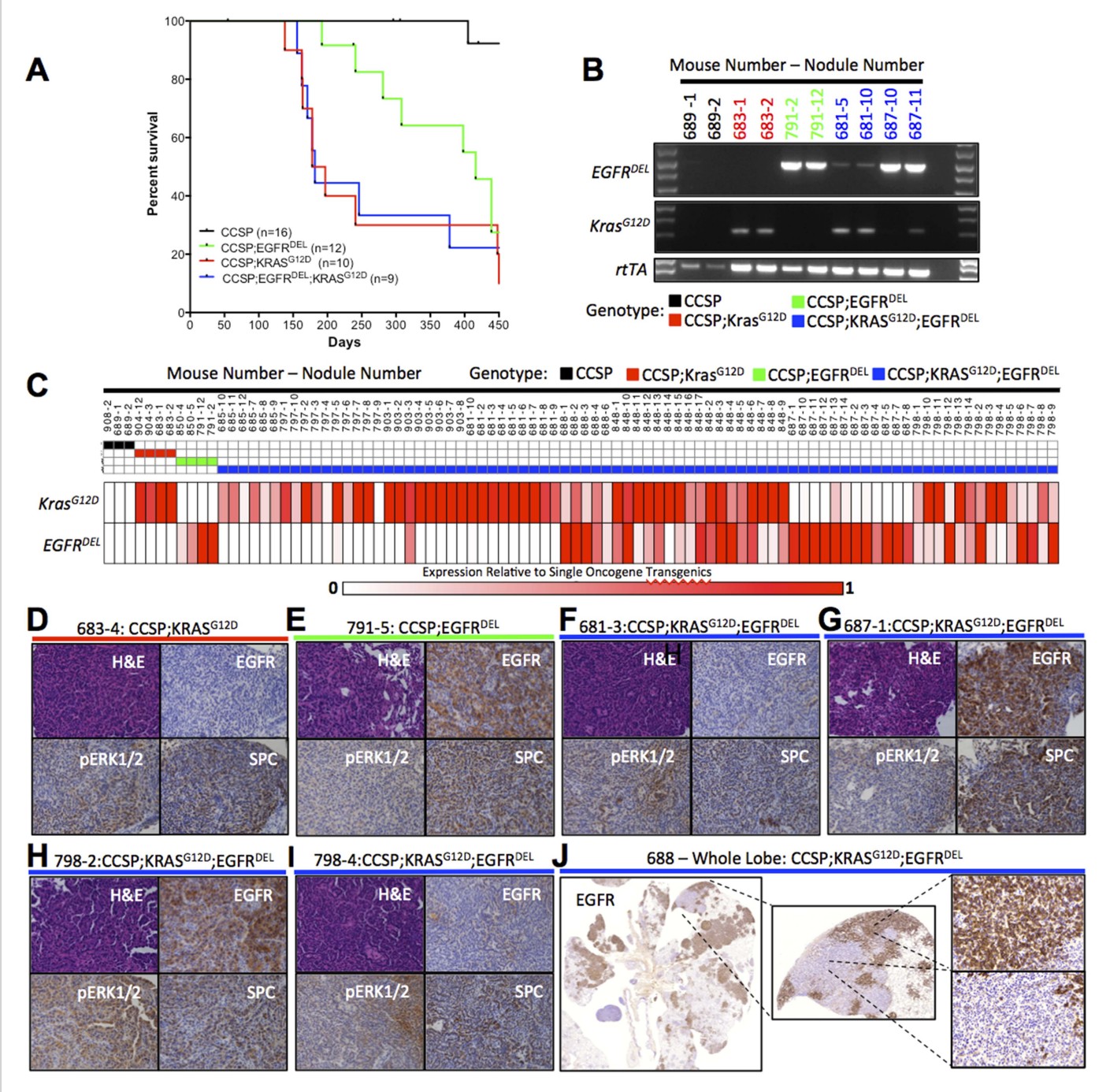

**Figure 2**. Co-induction of mutant Kras and mutant EGFR in the mouse lung epithelium leads to the development of LUAD that express a single oncogene. (**A**) Survival curves for mono-transgenic mice expressing only rtTA from the clara cell specific promoter (*CCSP*, black line); bi-transgenic mice expressing mutant *EGFR* (*CCSP;EGFR*$^{DEL}$, green line); bi-transgenic mice expressing mutant *Kras* (*CCSP;Kras*$^{G12D}$, red line); and tri-transgenic mice (*CCSP;EGFR*$^{DEL}$; *Kras*$^{G12D}$, blue line). (**B**) Levels of transgene-specific RNA from normal lung tissue (for CCSP only) and tumor nodules (for oncogene containing transgenics) as measured by gel electrophoresis of products of RT-PCR reactions primed by oligonucleotides specific for Kras$^{G12D}$, EGFR and rtTA RNA, prepared from animals encoding only rtTA (*CCSP*; black), Kras$^{G12D}$ (*CCSP;Kras*$^{G12D}$; red), EGFR$^{DEL}$ only (*CCSP;EGFR*$^{DEL}$;green) and EGFR$^{DEL}$ plus Kras$^{G12D}$ (*CCSP;EGFR*$^{DEL}$; *Kras*$^{G12D}$;blue). Mice are indicated by the initial number; the number following the dash represents the nodule or tissue number from that mouse. (**C**) EGFR$^{DEL}$ and *Kras*$^{G12D}$ RNA levels measured by qRT-PCR from normal lung (for *CCSP* mice only) and from tumor nodules (for all mice containing transgenic oncogenes). Data for each nodule are represented in a 'heat map' with the expression levels shown relative to the average of the respective bi-transgenic animals for each transgene (see 'Materials and methods'). Mice and nodules are identified as above. (**D–I**) Histological appearance (H&E staining) and IHC detection of relevant proteings (human EGFR, phospho-ERK1/2 and SPC) in selected mouse lung nodules also examined in panel **C**. *Figure 2. continued on next page*

Figure 2. Continued

Mouse and nodule numbers are indicated as before along with the respective genotype for each mouse; the genotype is also indicated by the color of the line below each label by the color scheme for genotype according to the key used in panel **C**. (**J**) IHC for human EGFR in a lung lobe from a tri-transgenic mouse, indicating regions of distinct staining within the same region of tumor.

redundant. However, the majority of tumors found in the tri-transgenic animals expressed only one of the TetO-oncogenes, as judged by RT-PCR (*Figure 2B*). Moreover, the levels of oncogenic RNA in tumors from tri-transgenic mice were similar to those observed in tumors from their bi-transgenic littermates. When RNA from both oncogenic transgenes was detected in occasional samples, the relative amounts varied, suggesting that the samples contained more than one tumor clone or adjacent non-transformed cells, as examined in greater detail below.

The findings in *Figure 2* could be explained by a low frequency of dual induction of expression of two oncogenic transgenes in lung cells. However, our laboratory has previously demonstrated in a mouse model of breast cancer that two oncogenic transgenes, *TetO-Kras^{G12D}* and *TetO-Myc,* can be co-expressed in mammary tumors at levels similar to those observed in tumors from transgenic animals containing only one of the transgenes (*Podsypanina et al., 2008*). Furthermore, in other tri-transgenic mice carrying the same *CCSP-rtTA* transgene, most if not all tumors expressing the oncogenic *TetO-EGFR^{L858R}* transgene also expressed a *TetO-Cre* transgene, as judged by elimination of a floxed *Erbb3* allele (*Song et al., 2015*). These findings suggest that our observations here reflect the incompatibility of mutant Kras and EGFR and not an inability to express both doxycycline-dependent transgenes.

Since *TetO-Kras^{G12D}* is the more potent oncogenic transgene in our mouse model, we supposed that if only one oncogenic transgene were expressed it would be the *Kras* transgene. But we initially found that two lung tumors from one tri-transgenic mouse preferentially expressed *Kras* and two tumors from a different tri-transgenic mouse expressed only *EGFR* (*Figure 2B*). Expanding the use of the qRT-PCR assay to examine over 90 tumor nodules from 14 mice, 8 of which were tri-transgenics, confirmed that tumors from tri-transgenic mice almost always contained RNA from a single transgenic oncogene, either *EGFR* or *Kras*, and the amounts of RNA were similar to those found in tumors from the bi-transgenic animals (*Figure 2C*). In some tri-transgenic mice (e.g., mouse number 687), all nodules contained only *EGFR* transgenic RNA while nodules from some others expressed only *Kras* transgenic RNA (e.g., mouse number 685). Furthermore, there were animals in which a different transgene, *Kras* or *EGFR,* was expressed in individual tumor nodules (e.g., mouse number 798), implying that tumors from mice carrying oncogenic alleles of both *Kras* and *EGFR* have evolved by selecting against cells that express both mutant transgenes, further suggesting that expression of both these genes is detrimental.

As shown in *Figure 2*, some tumor nodules appear to contain both *Kras* and *EGFR* transgenic RNAs, but we could not determine unequivocally whether this means that a few tumors expressed both oncogenes or that multiple tumors may have grown too closely together to separate during macrodissection. To address the latter possibility, we used immunohistochemistry (IHC) to ask whether *EGFR* and *Kras* transgenes are expressed in the same cells by assessing levels of human EGFR and by measuring phosphorylated ERK (p-ERK) as readout of MAP kinase (MAPK) signaling. This analysis, however, cannot definitely address co-expression as the intensity of p-ERK signaling may be enhanced by induction of either transgene alone. Tumors from bi-transgenic animals with the *EGFR* transgene scored positive for both human EGFR and p-ERK (*Figure 2E*), while tumors from bi-transgenic *Kras* animals scored positive only for p-ERK (*Figure 2D*). Some individual tumor nodules from tri-transgenic animals contained human EGFR and were positive for p-ERK (*Figure 2G*), but nodules that appeared to lack human EGFR scored positive for p-ERK, implying that they express only the *Kras* oncogene (*Figure 2F*). Furthermore, we confirmed that individual tumor nodules from one tri-transgenic animal might or might not contain human EGFR (*Figure 2H*) and if they did not, they still contained p-ERK, implying expression of *Kras* (*Figure 2I*).

Finally, an entire lung lobe from a tri-transgenic animal revealed adjacent regions of adenocarcinoma with different patterns of human EGFR staining, suggesting that distinct cell populations expressed either *Kras* or *EGFR* (*Figure 2J*). The proximity of these regions likely explains the detection of both transgenic RNAs by qRT-PCR in some instances. Together, the data from tri-transgenic animals suggest negative selection against co-expression.

## Co-expression of mutant KRAS and EGFR in cultured human lung adenocarcinoma cells decreases cellular viability

The findings with transgenic mice bearing inducible *Kras* and *EGFR* oncogenes imply that co-production of mutant EGFR and mutant KRAS is not tolerated in the same cell. To confirm and further explore this conclusion, we directly induced the expression of mutant *KRAS* or mutant *EGFR* in established lines of human lung adenocarcinoma cells known to be driven by either mutant EGFR (PC9 cells; in-frame deletion in exon 19) or by mutant KRAS (H358 cells; G12C) (*Arao et al., 2004*; *Sunaga et al., 2011*). For this purpose, we cloned mutant *KRAS* (G12V), mutant *EGFR* (L858R), and (as a control) *GFP* into doxycycline-inducible vectors (pInducer, [*Meerbrey et al., 2011*]) and established stable cell lines that express mutant *KRAS* or *GFP* in PC9 cells and mutant *EGFR* or *GFP* in H358 cells in response to doxycycline (Dox). In this single-vector system, a tetracycline-responsive promoter controls the genes while the tetracycline transactivator, rtTA, is expressed by the constitutive *Ubc* promoter (*Meerbrey et al., 2011*). Cells treated for 24 hr with Dox (100 ng/ml) produced the appropriate proteins, suggesting that this system could allow us to describe the consequences of oncogene co-expression (*Figure 3A*).

All four of the modified cell lines were maintained in the presence or absence of Dox for 7 days, and cell viability assessed on multiple days by incubating with the vital dye, alamar blue (*Figure 3B*). By day 7, there was a marked decrease in viable PC9 or H358 cells producing the additional mutant oncoprotein. Cells that produce GFP after exposure to Dox were not affected in this assay (*Figure 3B, C*). As a result, the numbers of cells in these cultures were significantly decreased by 7 days after induction (*Figure 3C*). These results support the hypothesis that co-expression of mutant EGFR and mutant KRAS is incompatible with cell survival.

To determine whether the toxic effects of the co-production of mutant EGFR and mutant KRAS depend on the tyrosine kinase activity of EGFR, we added erlotinib, an EGFR TKI used to treat human LUAD (*Pao et al., 2004*), to PC9 cells induced to express either the KRAS mutant or GFP. In the absence of mutant KRAS, erlotinib markedly decreased the number of PC9 cells, regardless of whether GFP was induced by Dox. However, erlotinib significantly protected PC9 cells from the toxic effects of induced mutant KRAS expression at day 7 (*Figure 3D*). Thus, the induction of mutant *KRAS* (but not induction of *GFP*) rescues PC9 cells from the lethal effects of erlotinib, implying that the toxicity of co-expression of mutant *KRAS* and *EGFR* depend on the kinase activity of mutant EGFR. In addition, these findings confirm the expected ability of mutant RAS to render EGFR-mutant adenocarcinoma cells resistant to TKI's and further enhance the significance of not observing oncogenic *KRAS* mutations in human tumors resistant to erlotinib (*Ohashi et al., 2012*) (see 'Discussion').

## How does co-expression of mutant KRAS and mutant EGFR cause cell toxicity?

Having established that co-expression of mutant *KRAS* and *EGFR* oncogenes affects cell survival, we asked whether known mechanisms of cell death, such apoptosis and autophagy, were responsible for the observed consequences of co-expression. On day 5 after induction of the second oncogene in either cell line, both H358 and PC9 cells showed reduced viability and exhibited increases in common indicators of apoptosis, cell surface staining for Annexin V and 7-AAD permeability (*Figure 4A*). The increase in these markers of apoptosis was more evident in PC9 cells producing mutant KRAS than in H358 cells producing mutant EGFR. This was also true when measuring PARP cleavage, a marker of caspase activity (*Figure 4B*). This suggests that conventional apoptosis may play a role in the decreased viability in both scenarios of oncogene co-expression, but is more pronounced in the modified PC9 than in H358 cells.

Autophagy is a process whereby intracellular proteins and whole organelles are catabolized in a special double-membraned structure termed the autophagosome. A key component of the autophagosome is LC3, light-chain 3, a microtubule-associated protein that is uniquely modified by phosphatidylethanolamine (LC3-II) (reviewed in [*Mizushima, 2007*]). We observed an increase in the modified LC3-II (the faster migrating protein in the gels shown in *Figure 4C*) at day 5 in PC9s and to a lesser, but appreciable extent in H358 cells. This suggests that autophagy may also be involved in the loss of cell viability in these cells.

To further characterize the effect of expressing both mutant KRAS and mutant EGFR, we followed cell cycle progression by measuring DNA content by propidium iodide staining. In H358 cells producing mutant EGFR, we did not observe a major difference in cell cycle stages (G1, S, or G2/M) at

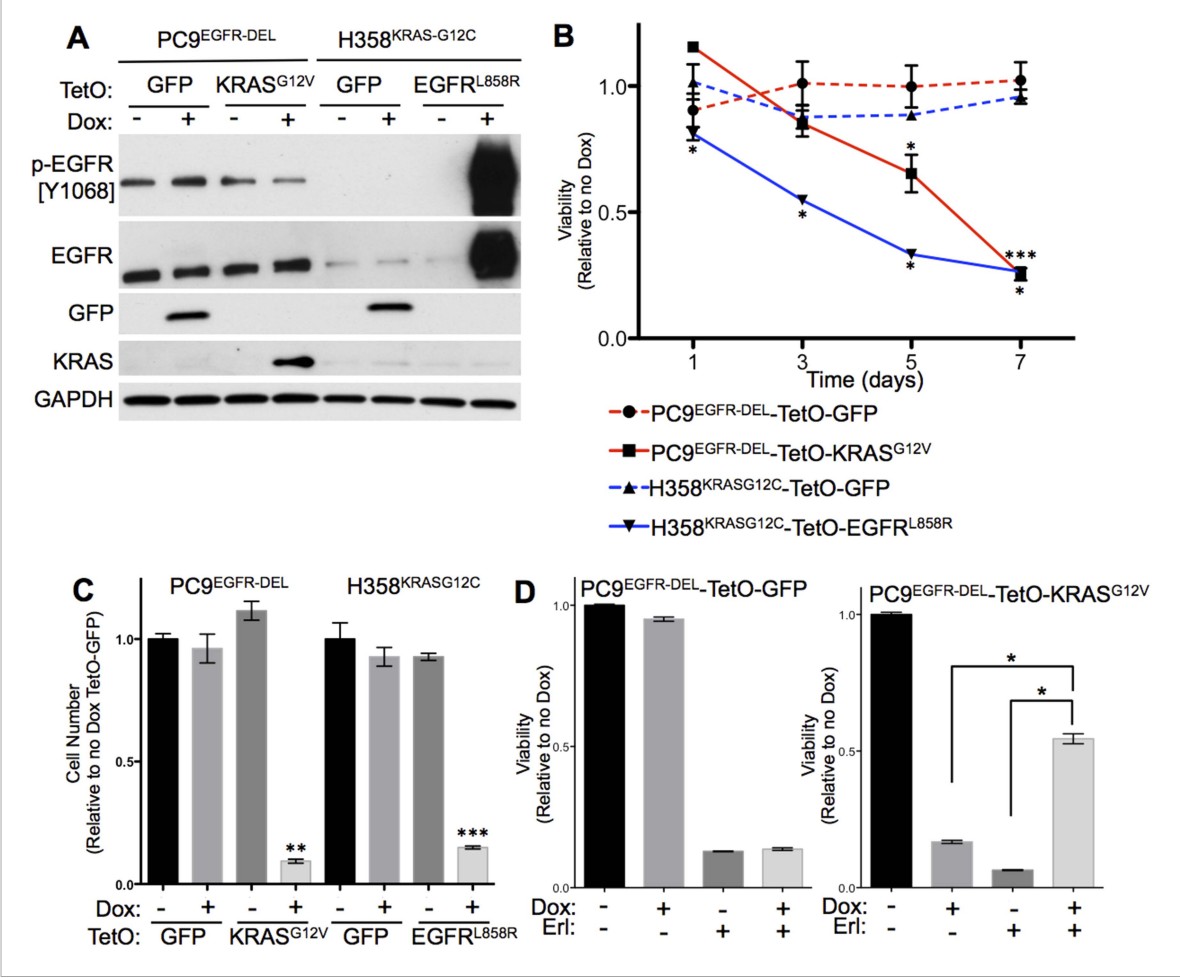

**Figure 3**. Co-expression of mutant KRAS and mutant EGFR decreases viability of human lung adenocarcinoma cell lines. (**A**) Induced expression of transduced genes in human lung cancer cell lines. PC9 and H358 cell lines were transduced with the indicated tetracycline-responsive plasmids. Lysates of cells cultured in the presence or absence of doxycycline (Dox) for 24 hr were prepared and assayed for the indicated protein expression by western blotting, as described in Materials and Methods. (**B**) Co-expression of mutant KRAS and EGFR reduces cell viability. Cells were grown in the presence or absence of Dox for up to 7 days and tested for viability by alamar blue. Averaged values from three independent experiments were normalized and plotted for each cell line relative to untreated cells (no Dox) at the indicated time points. Error bars represent ± standard deviation (SD) for each point. p-values were calculated between the + and − Dox states of individual cell lines at each time point using a two-tailed, unpaired t test with Welch's correction. *, ** and *** represent significance values <0.01, <0.001 and <0.0001, respectively. (**C**) Co-expression of mutant *KRAS* and *EGFR* reduces cell number. Cells were grown as in panel **B** and counted on day 7. Average cell number from three independent experiments were normalized and plotted for each cell line relative to cells expressing *TetO-GFP* in the absence of Dox. Error bars represent ± standard error of the mean (SEM) and p-values were calculated between the + and − Dox states as described in **A**. (**D**) Erlotinib protects PC9-*TetO-KRASG12V* cells from the toxic effects of oncogene co-expression. Cells were grown with or without erlotinib (Erl) and/or Dox for 7 days and cell viability determined by alamar blue. Results from three independent experiments were normalized and plotted for each cell line relative to untreated cells (no Dox) with error bars representing ± SEM. p-values were calculated between the groups indicated as in **A**.

days 3 or 5 after Dox addition (*Figure 4D*). However, in PC9 cells producing mutant KRAS, we found an approximately twofold increase in cells at G2/M treated with Dox (*Figure 4D*). Thus, to varying levels, oncogene co-expression affects apoptosis, autophagy and cell cycle. Some combination of these processes and those described below, contribute to cell toxicity.

## Co-expression of mutant KRAS and EGFR induces morphological changes in lung adenocarcinoma cells

During the course of these studies, we observed distinctive changes in cell morphology after doxycyline-mediated induction of a second oncogene in PC9 and H358 lung cancer cell lines. Both

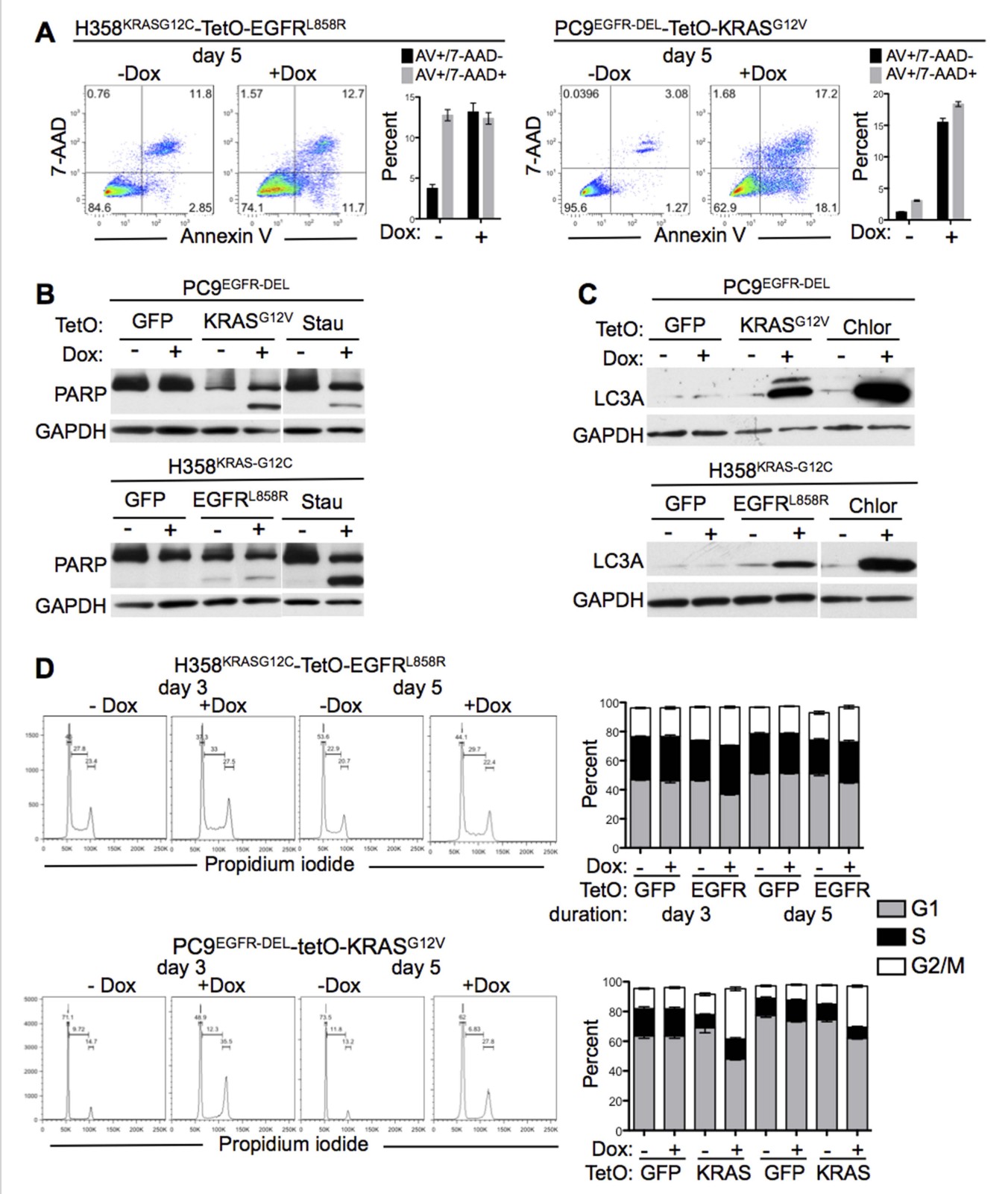

**Figure 4**. Effects of mutant KRAS and EGFR co-expression. (**A**) Measurement of apoptosis after co-expression of mutant oncogenes. The indicated cells were grown for 5 days with or without Dox and assessed for apoptosis by flow cytometry for Annexin V and 7-AAD. Data are plotted as either Annexin V positive/7-AAD negative (AV+/7-AAD−, early apoptosis, black bars) or Annexin V/7-AAD positive (AV/7-AAD+, late apoptosis, gray bars) cells. (**B**) PARP

*Figure 4. continued on next page*

*Figure 4. Continued*

cleavage after co-expression of mutant oncogenes. Cells were grown as in panel **A** (5 days) and lysates assayed for cleavage of PARP protein by western blotting. Cells treated with staurosporine (Stau) for 24 hr were used as a positive control. (**C**) LC3A lipidation with co-expression of mutant oncogenes. Cells were grown as above (5 days) and assayed for autophagy by measuring the levels of LC3-II, the faster migrating band of LC3 (where applicable). Cells treated with chloroquine (Chlor) for 16 hr were used as a positive control. (**D**) Cell cycle analysis in cells co-expressing mutant oncogenes. Cells were grown for 3 and 5 days as indicated after induction of a second oncogene, and cell cycle status was determined by propidium iodide staining and flow cytometry. The fraction of cells in G1, S and G2/M are gated as indicated. The percentages of cells in each cell cycle stage are plotted as stacked bars. Data from **A** and **D** are plotted as averages from cells grown in triplicate wells ± SEM all data are representative of multiple independent experiments.

H358-TetO-EGFR[L858R] and PC9-TetO-KRAS[G12V] cells displayed ruffling of the cell membrane and large phase-translucent vacuoles (*Figure 5A*), both more prominent with time. These features are similar to what has been described in glioblastoma cell lines undergoing a process recently termed methuosis (*Chi et al., 1999*; *Overmeyer et al., 2008*) (see 'Discussion'). To further explore the morphological changes in cells co-expressing EGFR and KRAS oncogenes, we observed the cells by transmission electron microscopy (EM) (*Figure 5B*). The presence of large, single membrane, vacuoles were detectable in cells induced to express a second mutant oncogene. Microvilli were often present on the vacuolar membrane of these cells (*Figure 5B*).

Prompted by our observations and published studies, we asked whether lung cancer cell lines expressing the two mutant oncogenes displayed features of macropinocytosis (*Overmeyer et al., 2008*; *Commisso et al., 2013*; *Kitambi et al., 2014*). 3 days after Dox addition, PC9-TetO-KRAS[G12V] and H358-TetO-EGFR[L858R] cells were incubated with fluorescent 70 kDa dextran to gauge their capacity to take up this macromolecule. Based on flow cytometry 30 min after incubation, both cell lines co-expressing the two oncogenes showed increased macropinocytosis relative to the untreated lines that naturally express mutant KRAS or mutant EGFR but not both (*Figure 5C*). Thus, uncontrolled fluid phase uptake may be one of the pathological consequences of oncogene co-expression.

## Co-expression of mutant KRAS and EGFR increases MAPK signaling lung adenocarcinoma cells

To identify signaling pathways that may be responsible for the decreased cell viability and morphological changes induced by co-expression of mutant *KRAS* and *EGFR*, we generated gene expression profiles of the PC9 and H358 lung adenocarcinoma cells engineered to conditionally express mutant *KRAS* or *EGFR*. 24 hr after addition of Dox to PC9-*TetO-KRAS[G12V]* and H358-*TetO-EGFR[L858R]* cells or to control lines with *TetO-GFP* we harvested RNA. Gene expression profiles were then compared to profiles from untreated cells. The 24 hr time point was chosen to measure changes in cell signaling likely to be caused directly by induction of the co-expressed oncogene rather than by reactive changes that could be attributed to secondary events occurring at later time points. Genes differentially expressed in cells treated or not treated with Dox were identified for each cell line. Those genes showing significant differences (ANOVA Corrected $p < 0.01$, compared to reciprocal no Dox control) were examined in similar tests with cells induced to express *TetO-GFP* to determine those specifically affected by co-expression of the two oncogenes ('Materials and methods'). In total, 152 probe sets corresponding to 144 unique genes detected differential expression in both H358 and PC9 cells in response to mutant EGFR and KRAS (*Figure 6A*, *Supplementary file 3*).

To determine the signaling pathways likely to be activated based on the differential RNA profiles, we performed Gene Set Enrichment Analysis (GSEA) (*Subramanian et al., 2005*) and observed that the most significant enrichment occurred for RNAs encoding downstream targets of oncogenic KRAS (*Figure 6B*, *Supplementary file 4*). Furthermore, Ingenuity Pathway Analysis (IPA) revealed p38 MAPK and ERK/MAPK signaling as the most significantly up-regulated canonical pathways (*Figure 6C*). As MAPK signaling is immediately downstream of both EGFR and KRAS (*Figure 6D*), we asked whether further activation of this pathway might occur in response to expression of both mutant oncogenes.

To gain a better understanding of EGFR-RAS-MAPK signaling in the two cell lines, we assayed molecules in this pathway for phosphorylation status as a surrogate for signaling activity (*Figure 6D*). 3 days after the addition of Dox, EGFR was constitutively phosphorylated in PC9 cells, as expected,

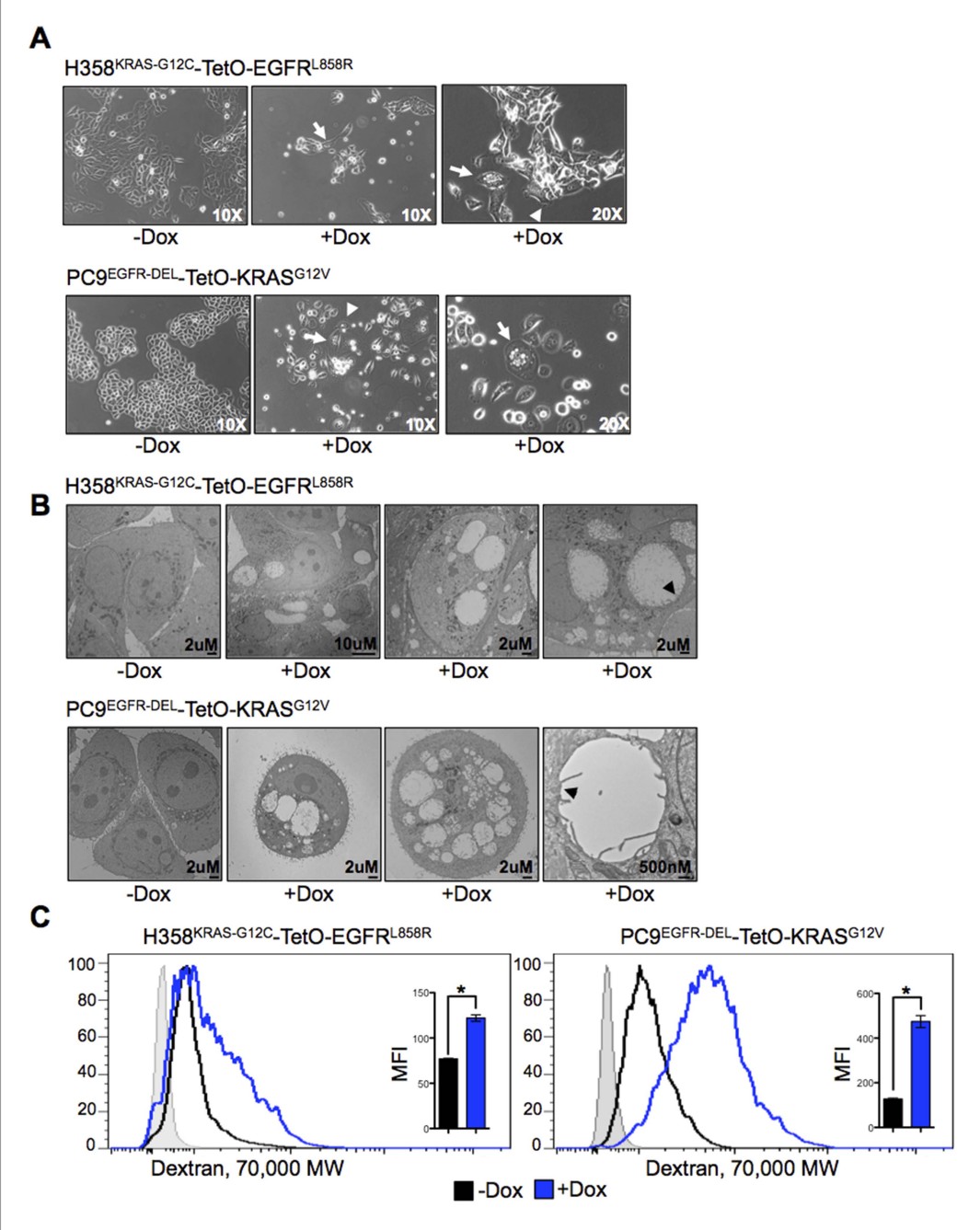

**Figure 5**. Co-expression of mutant KRAS and EGFR induces morphological changes and increased macropinocytosis in lung adenocarcinoma cells. (**A**) Changes in morphology induced by co-expression of mutant oncogenes. Cells were grown for 5 days after induction of expression of a second oncogene, and phase contrast images were taken with indicated objectives. Arrows show a vacuolated cell and arrowheads show membrane ruffling. Images are representative of cell morphology during the analyses (as early as day 1 and as late as day 7 after Dox). (**B**) Vacuolization in cells co-expressing mutant oncogenes. Cells were cultured for 5 days after induction of a second oncogene and analyzed by transmission EM (TEM). Scale bars on lower right. Far right panels for each cell type shows a vacuolar structure bound by a single membrane (black arrowhead). (**C**) Increased macropinocytosis with co-expression of mutant oncogenes. Cells were cultured for 3 days in the presence or absence of Dox and assayed for macropinocytosis by uptake of fluorescent 70 kDa Dextran for 30 min. Dextran uptake was measured by flow cytometry. A histogram of data from one experiment and the average MFI (median fluorescence intensity) from three independent experiments ± SEM are also shown. The shaded light gray histogram represents fluorescence intensity of cells that received neither Dextran nor Dox. p-values were calculated between the + and − Dox states using a two-tailed, unpaired t test with Welch's correction with * representing significance values <0.01.

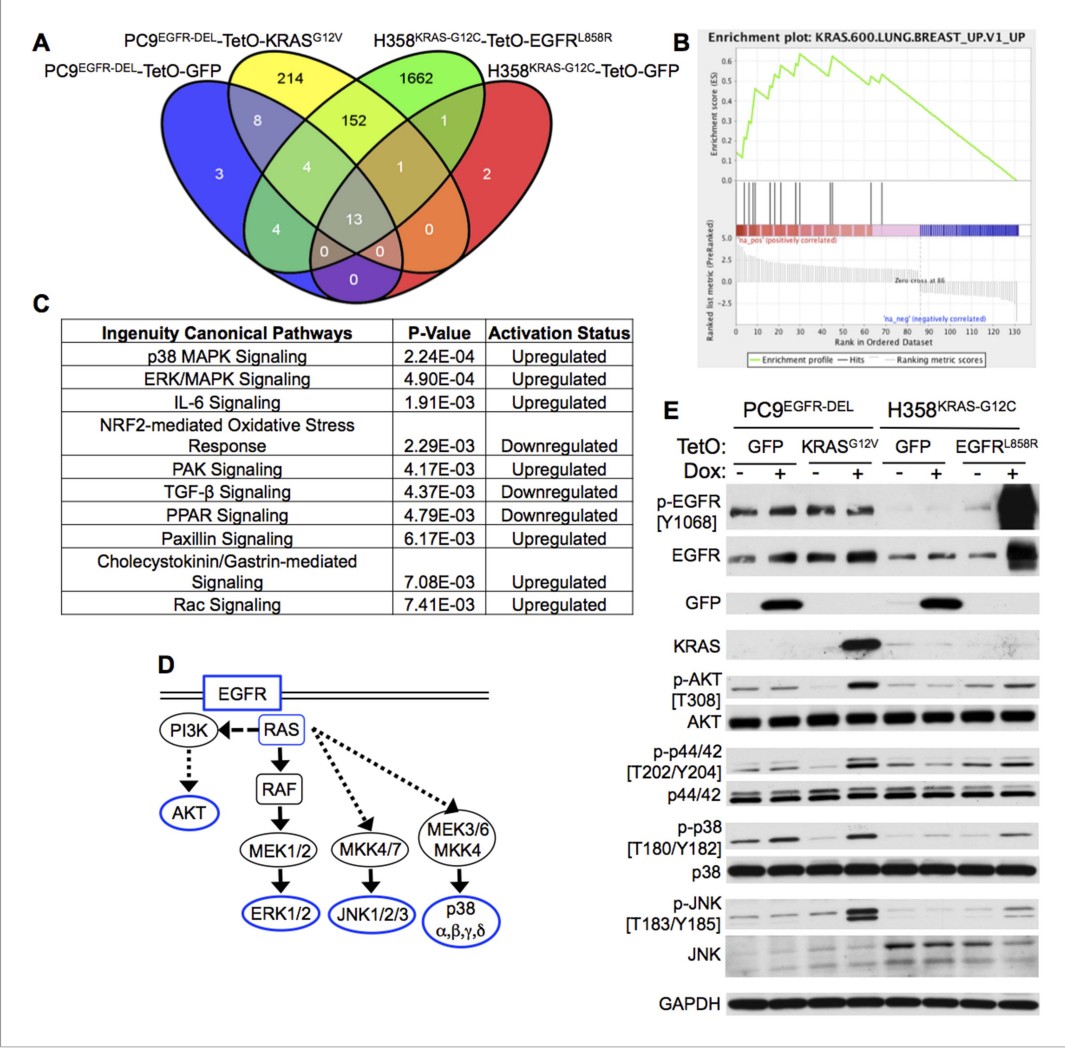

**Figure 6**. Co-expression of mutant KRAS and EGFR increases MAP kinase (MAPK) signaling. (**A**) Modified PC9 and H358 lung adenocarcinoma cells (see *Figure 3*) were cultured in the presence or absence of doxycycline (Dox) for 24 hr and analyzed for global gene expression changes using Affymetrix microarrays (see 'Materials and methods' for details). Microarray probes differentially expressed in each TetO cell line upon the addition of Dox were identified (corrected p < 0.01, compared to control without Dox), and those genes specifically induced by Dox in either TetO-KRAS or TetO-EGFR cells and not the TetO-GFP control cells were determined. The Venn diagram indicates the resulting number of gene probes identified in each cell line, including the 152 unique probes specifically modulated by expression of mutant *KRAS* and *EGFR* in both PC9 and H358 cells. (**B**) Gene Set Enrichment Analysis (GSEA) of the genes specifically regulated upon mutant *KRAS* and *EGFR* co-expression in both PC9 and H358 lung adenocarcinoma cells identified three oncogenic signatures that were significantly upregulated (FDR q-value <0.01) upon co-expression; the top two are indicative of KRAS signaling (see 'Materials and methods' and *Supplementary file 5*). The displayed enrichment plot is for the most significant gene set (q-value = 0, Normalized Enrichment Score = 2.29) demonstrating enrichment for genes related to the upregulation of mutant *KRAS*. (**C**) Ingenuity Pathway Analysis (IPA) of the KRAS + EGFR induced gene set was performed and the top ten significantly regulated canonical pathways in which these genes are involved are displayed (see 'Materials and methods'). P38 MAPK signaling was identified as the most significant upregulated pathway from this analysis; ERK/MAPK signaling was the second. (**D**) A highly simplified diagram of the EGFR/RAS signaling pathway is illustrated; the components assessed by western blot highlighted in blue. (**E**) Increased MAPK signaling in cells co-expressing mutant oncogenes. The indicated cells were cultured for 3 days with or without Dox; lysates were assayed by western blotting for the indicated proteins and phospho-proteins. Where relevant, the phosphorylated Tyrosine (Y) or Threonine (T) residue being measured is shown. Data are representative of three independent experiments.

whereas EGFR was not appreciably phosphorylated in H358 cells, except when mutant *EGFR* was induced (*Figure 6E*). Phosphorylation of AKT at T308 was strongly increased in PC9 cells expressing mutant *KRAS^{G12V}* but less so in H358 cells expressing mutant *EGFR*. We observed a similar pattern for phospho-ERK (p44/42) and phospho-p38, the latter not being associated with normal EGFR signaling (*Figure 6E*). The most consistent change in both cell lines was the phosphorylation of JNK, a stress-activated kinase. The activation of JNK can occur through the Rho family of GTPases, and the Rho protein family has also been implicated in RAS-dependent increases in macropinocytosis (*Bhanot et al., 2010*). The kinase that phosphorylates both JNK and p38, MKK4, has been shown to be activated by the small molecule vacquinol-1, an inducer of macropinocytosis (*Kitambi et al., 2014*); this is discussed further below.

Taken together, our data suggest a general mechanism that is similar to the collective observations of other groups: increased activity of p38 and JNK MAPK pathways, increased macropinocytosis, and vacuolization lead to cell death. We show here that this cellular state can be created by the co-incidental activity of two mutant oncogenes like *KRAS* and *EGFR*. Thus, our data could provide an explanation for the observed mutual exclusivity.

## Discussion

Our findings argue strongly that the well-known mutual exclusivity of *KRAS* and *EGFR* mutations in LUAD is due to synthetic lethality of the two mutant oncogenes, rather than to functional redundancy. Because multiple rounds of mutation are followed by clonal selection during tumor evolution, selection against cells with lethal combinations of mutations will produce mutual exclusivity. This general concept could have important implications for the design of therapeutic strategies, as well as tumor evolution. Recent efforts to design new treatments have focused on mutant genes that, when inhibited, decrease the viability of cancer cells (*Sawyers, 2009*). In contrast, combinations of mutations that are not found in tumor types may suggest synthetically lethal effects that could be exploited therapeutically—for instance, by using signaling agonists to provoke events that are lethal only in the presence of another mutant oncogene.

In this study, we focused on one synthetically lethal relationship—oncogenic mutations of *KRAS* and *EGFR* in LUAD. We have used three approaches to ascertain the relationship.

i. First, by analyzing available DNA sequences from large numbers of LUAD, we confirmed that activating mutations in these two genes are mutually exclusive, regardless of the smoking history of patients (*Figure 1A*). Furthermore, we found that this kind of relationship is rare. No other mutated pairs of genes demonstrated a similarly negative association in lung adenocarcinoma, further arguing that the mutual exclusivity of *KRAS* and *EGFR* mutations is due to negative selection. By applying the same 'exclusivity analysis' to mutation data from other tumor types, however, we identified or confirmed other pairs of mutations—mutant *KRAS* and *BRAF* in COAD, mutant *EGFR* and *NF1* in GBM, and mutant *BRAF* and *NRAS* in SKCM (*Petti et al., 2006*; *Sensi et al., 2006*)—that may be synthetically lethal in those tumors. Additionally, this analysis confirms previous co-operative mutations (e.g., *KRAS* and *LKB1*) and reveals potentially new, testable gene pairs that are selected for during tumor formation (*Figure 1*). However, we note that this analysis was confined to alleles with altered sequences in coding domains and did not include gene rearrangements and amplifications. Thus we cannot draw conclusions about whether certain kinds of combinations, such as amplification of one proto-oncogene and sequence mutation of another, create synthetic lethalities. Moreover, the data sets we examined were derived by profiling cancers from patients who had not received medical treatment; hence the relationship of drugs or drug resistance to synthetic lethality has not been examined, and we would not have detected the recently reported amplification of wild type *KRAS* in a COAD with a mutant *BRAF* gene (*Ahronian et al., 2015*).

ii. Second, we obtained strong support for the concept of synthetic lethality of mutant KRAS and EGFF proteins in LUAD by attempting to co-express doxycycline-inducible transgenes encoding the two proteins in mouse models. Although each oncogenic transgene could initiate lung tumorigenesis on its own (*Fisher et al., 2001*; *Politi et al., 2006*), if both were present in the same mouse, the resulting tumor nodules expressed only one of the two transgenes (*Figure 2*); this implies selection against cells that maintain expression of both.

iii. Finally, to observe the proposed synthetic lethality more directly, we engineered an inducible second oncogene into human lung adenocarcinoma cell lines carrying a mutant *EGFR* or *KRAS* gene (*Figures 3–6*). We found that expression of the introduced second oncogene was detrimental in cells expressing both oncogenes.

By taking advantage of cell lines that can be induced to express the second oncogene in a lethal combination, we have been able to study the manner and mechanisms of cell death in this context. We were motivated to pursue these issues to seek insight into the factors governing the synthetic lethality of mutant EGFR and KRAS and to identify opportunities to exploit this knowledge for therapeutic benefit. We found that the expression of both oncogenes caused the death of lung cancer cells within a week, mostly in a non-canonical manner. While we observed some markers of classical apoptosis and autophagy and small changes in cell cycle distribution in one line, the dominant features were cell enlargement and vacuolization, activation of stress-associated protein kinases (as judged by phosphorylation of JNK and p38 MAPK), and increased macropinocytosis, with a subsequent loss of cell viability.

Macropinocytosis and vacuolization have been observed previously in cells with augmented RAS signaling, nutrient acquisition, and cell death (*Chi et al., 1999*; *Overmeyer et al., 2008*; *Commisso et al., 2013*), although not as an explanation of synthetic lethality caused by multiple oncogenes. Expression of H-Ras$^{G12V}$ in glioblastoma and gastric cancer cell lines generates vacuoles associated with increased cell death (*Chi et al., 1999*), a phenomenon dubbed 'methuosis' (*Maltese and Overmeyer, 2014*). These findings were subsequently linked to induction of uncontrolled macropinocytosis though activation of the Rac1 GTP-binding protein (*Bhanot et al., 2010*). More recently, a small molecule screen for inhibitors of glioblastoma cell lines identified a quinine derivative (vacquinol-1) that induced cell vacuoles, membrane ruffling, and increased macropinocytosis, similar to the effects of RAS activation alone (*Kitambi et al., 2014*). This process was found to be dependent on MKK4, part of the MAP2K family, that can phosphorylate both p38 and JNK (*Kitambi et al., 2014*). As Rac1 is known to activate both p38 and JNK kinase pathways (*Coso et al., 1995*; *Minden et al., 1995*; *Olson et al., 1995*) these findings imply the existence of a common mechanism whereby RAS-mediated activation of Rac1 stimulates p38 and JNK, prompting uncontrolled macropinocytosis, vacuolization, and subsequent cell death. Despite these supportive studies, however, the findings that implicate JNK and p38 pathways in the mechanism of synthetic lethality induced by mutant KRAS and EGFR co-expression are only correlative at this point. Additional studies will be needed to determine whether activation of one or both of these pathways is required for cell death and whether it causes or results from macropinocytosis and vacuolization.

Parts of this scenario, however, may not always be detrimental. Commisso et al. have shown that KRAS$^{G12V}$ induces macropinocytosis unaccompanied by vacuolization and cell death (*Commisso et al., 2013*), suggesting that macropinocytosis may augment amino acid uptake, as required by cells growing rapidly under the influence of RAS signaling. In the context of active EGFR and KRAS, it is possible that both mutant alleles together increase RAS-MAPK-mediated signaling beyond a tolerable threshold, resulting in increased activation of Rac1 and subsequent stimulation of p38 and JNK, kinases that are not typically activated to high levels by mutant KRAS or EGFR alone. Indeed, our data demonstrate increased activation of KRAS signaling and activation of stress kinases p38 and JNK upon co-induction of mutant KRAS and EGFR. Furthermore, these signaling events may be accompanied by, uncontrolled macropinocytosis, resulting in the accumulation of macropinosomes in the cytosol that overwhelm the capacity of the cells to adapt to the presence of large vacuoles. Thus, unlike a situation in which macropinocytosis favors cell survival, this would result in cell death. Future work will be required to determine the biochemical details of this process, such as the role of Rac1, and to test whether the observed mutual exclusivity of other combinations of mutant genes in other tumor types are attributable to such mechanisms.

Our findings also have potential implications for understanding mechanisms of resistance to EGFR-targeted therapies in lung adenocarcinoma. While inhibitors of EGFR have produced dramatic responses when used to treat EGFR-mutant tumors, acquired resistance to these compounds inevitably emerges (reviewed in [*Chong and Janne, 2013*]). Tumor cells acquire resistance to these compounds through multiple mechanisms, including a second-site mutation (T790M) in *EGFR*, amplification of *MET* or *ERBB2,* and conversion to a small cell lung cancer phenotype (*Yu et al., 2013*). Our findings and those from other groups (*Sharifnia et al., 2014*) demonstrate that mutant *KRAS* can also confer resistance to TKIs in *EGFR* mutant cell lines. Yet mutations in *KRAS* have not been found to explain acquired resistance to EGFR TKIs in patients (*Ohashi et al., 2012*; *Yu et al., 2013*). Synthetic lethality could explain this observation. Acquired resistance is generally caused by the selection of a minor clone initially present in a heterogenous tumor, in which the resistance mutation exists in a subset of cells prior to TKI treatment. However, if expression of both mutant *EGFR* and mutant *KRAS*

is detrimental, co-mutated cells are unlikely to survive during tumor evolution; therefore, they are unlikely to explain TKI resistance in an *EGFR*-mutant tumor.

Taken together, our findings offer convincing evidence that the mutual exclusivity of mutant KRAS and EGFR in lung adenocarcinoma is dictated by synthetic lethality. These findings may have major implications for the future development of new agonistic treatment strategies for a substantial fraction of lung cancers driven by mutant KRAS or EGFR.

## Materials and methods

### Sequence data and exclusivity analyses

To investigate the association between KRAS and EGFR mutations in human LUAD, sequence data was obtained from four different sources (*Ding et al., 2008*; *Imielinski et al., 2012*; *Seo et al., 2012*; *Cancer Genome Atlas Research Network, 2014*). All studies assessed the entire coding region of these genes in tumors and in matched normal specimens, and they provide information regarding the somatic mutation status (single nucleotide changes, small insertions and deletions) for 662 LUAD in total. Data for three of these studies (*Ding et al., 2008*; *Imielinski et al., 2012*; *Cancer Genome Atlas Research Network, 2014*) were downloaded from the cBioPortal for Cancer Genomics under three headings: 'Lung Adenocarcinoma (Broad, Cell 2012)', 'Lung Adenocarcinoma (TSP, Nature 2008)' and 'Lung Adenocarcinoma (TCGA, Provisional)' (*Cerami et al., 2012*; *Gao et al., 2013*). Data from Seo et al. was obtained from *Supplementary file 3* (*Seo et al., 2012*). Somatic mutations were investigated for their potential impact on protein function and status in the Catalogue of Somatic Mutations in Cancer (COSMIC) using MutationAssessor as previously described (*Cancer Genome Atlas Research Network, 2011*). Mutations in *EGFR* and *KRAS* were classified as oncogenic if they were predicted to have a substantial impact on protein function (insertion/deletion mutations or a High/Medium MutationAssessor Predicted Functional Impact Score for non-synonymous mutations) or were previously reported in additional tumors in COSMIC. Smoking status was obtained for each dataset and samples classified as 'Smokers' or 'Never Smokers', based on the categorization used in each individual study. To assess the relationship between mutations in two genes, $2 \times 2$ Fisher's Exact Tests were computed using an online software tool (http://www.langsrud.com/fisher.htm). To predict a negative association between mutant genes, a left-tailed p-value (which assesses the negative association between variables) was computed; a value $\leq 0.01$ was considered significant.

For the exome-wide analyses, variant level data was downloaded from TCGA Data Portal (Version 0.4.0, 2013-07-16, https://tcga-data.nci.nih.gov/tcga/). Redundant samples were removed to yield data for 520 unique tumor samples. All variants not mapping to coding regions were removed, and the resulting data files were parsed to stratify samples according to non-silent mutation (insertions/deletions, non-synonymous nucleotide variants) status of a given proto-oncogene. The silent (synonymous nucleotide variant) and non-silent mutation status for all other genes was then determined separately on a sample by sample basis and compared to those samples with non-silent mutations to determine the number of times the gene is mutated in a tumor with or without the mutated proto-oncogene. The resulting values for each gene (the number of samples with mutations in a gene and the proto-oncogene, number of samples with mutation in the gene and not the proto-oncogene, number of samples without mutations in the gene and with mutation in proto-oncogene, and number of samples without mutation in either) were then compared using the Fisher's Exact Test function in R (http://www.r-project.org/). Both 'greater' and 'lesser' p-values were calculated in order to determine positive and negative associations with a p-value of $\leq 0.01$ considered significant. However, it should be noted that these analyses only take into account gene mutations (nucleotide level variants and small insertions/deletions) and not copy number or methylation changes which potentially affect gene function. All plots were created using Prism software (GraphPad, La Jolla, CA). The same analyses were repeated for COAD, GBM and SKCM using the processed MAF files described in *Kandoth et al. (2013)*.

### Mouse models

All animals were kept in specific pathogen-free housing with abundant food and water under guidelines approved by the NHGRI Institutional Animal Care and Use Committee. The mouse models used in this study have been previously described (*Fisher et al., 2001*; *Politi et al., 2006*). All mice have been re-derived and backcrossed onto a FVB/N background (*Taketo et al., 1991*) which are more prone to develop spontaneous lung tumors later in life (>14 months) than the previous mixed

background for these models (*Mahler et al., 1996*). Tail DNA was isolated using the DNeasy Blood & Tissue Kit (Qiagen, Venlo, Netherlands) according to the manufacturer's protocol. Detection of the rtTA activator, *Kras^{G12D}*, and *EGFR^{DEL}* transgenes was performed as described previously (*Fisher et al., 2001*; *Politi et al., 2006*). Bi-transgenic *CCSP-rtTA;TetO-Kras^{G12D}* mice were bred with bi-transgenic *CCSP-rtTA;TetO-EGFR^{DEL}* mice to generate experimental animals. Doxycycline was administered by feeding mice with doxycycline-impregnated food pellets (Harlan-Teklad 625 ppm, Indianapolis, IN). Mice were sacrificed when showing obvious signs of distress and lung tissues were processed and analyzed for the presences of lung tumors as described below. All survival curves were generated using Prism software (GraphPad), with only mice that presumably died from lung cancer indicated in the percent survival.

## Quantitative RT-PCR

Tissue samples were homogenized and RNA was extracted using the RNeasy Mini kit (Qiagen) according to the manufacturer's instructions. DNase I (Qiagen) treatment was performed to eliminate any contaminating transgene DNA. RT–PCR reactions were carried out using the High-Capacity cDNA Reverse Transcription Kit (Life Technologies, Carlsbad, CA). Custom TaqMan Gene Expression Assays (Life Technologies) were designed based on the sequences amplified by the genotyping primer sets (*Fisher et al., 2001*; *Politi et al., 2006*) and quantitative RT–PCR reactions were performed using standard TaqMan reagents and protocols on a 7900 HT Fast Real-Time PCR system (Life Technologies). The $\Delta\Delta Ct$ method was used for relative expression quantification using the average cycle threshold for B-actin RNA (Life Technologies Mm01324804_m1) to normalize gene expression levels between samples. Resulting fold change for each sample was compared against the average expression of bi-transgenic control mice and the values plotted using GENE-E software (http://www.broadinstitute.org/cancer/software/GENE-E/index.html).

## Histology and IHC

Animals were sacrificed with a lethal dose of $CO_2$ per institutional guidelines. The whole lung or lung nodules were excised and either flash-frozen or directly immersed in 4% paraformaldehyde in PBS (whole lungs). All tissues were fixed in 4% paraformaldehyde overnight at room temperature, placed in 70% ethanol, and sent for paraffin embedding and sectioning (Histoserv, Germantown, MD). Hematoxylin and eosin stain (H&E) were performed to assess histology and confirm the presence of tumors. The antibodies used for IHC were anti-human-EGFR K1492(Dako, Carpinteria, CA), anti-prosurfactant protein C AB3428(EMD Millipore, Billerica, MA), and anti-phospho- MAPK Thr202/Tyr204 4376 (Cell Signaling Technology, Danvers, MA) as previously described (*Fisher et al., 2001*; *Politi et al., 2006*).

## Cell lines and culture conditions

PC9 (PC-9) and H358 (NCI-H358) cells were obtained from Dr Romel Somwar at Memorial Sloan-Kettering Cancer Center. Cells were maintained in RPMI-1640 medium (ATCC, Manassas, VA) supplemented with 10% Tetracycline-free FBS (Clontech, Mountain View, CA) and 1% penicillin-streptomycin solution (Lonza, Basel, Switzerland). Cells were cultured at 37°; air; 95%; $CO_2$, 5%. Cell lines were authenticated by multiplex PCR (Genewiz, South Plainfield, NJ). Where indicated, doxycycline hyclate (Sigma-Aldrich, St. Louis, MO) was added at the time of cell seeding at 100 ng/ml. Erlotinib (Cell Signaling) was added at the time of cell seeding at 1 μM.

## Plasmids and generation of stable cell lines

Human mutant *EGFR* (Addgene plasmid #11012, [*Greulich et al., 2005*]), human mutant *KRAS* (Addgene plasmid #12544, [*Khosravi-Far et al., 1996*]) were subcloned into pENTR/D-TOPO (Life Technologies) and transferred by Gateway LR Clonase II enzyme mix (Life Technologies) to pInducer20 (gift from S Elledge, Harvard). Plasmids were sequence-verified. Lentivirus was generated using 293T cells (ATCC), psPAX2 #12260(Addgene, Cambridge, MA) and pMD2.G (Addgene plasmid #12259). Both polyclonal cell lines and single cell-derived clonal cell lines were used.

## Protein extractions and western blots

Cells were lysed in buffer (9803, Cell Signaling) containing complete protease/phosphatase inhibitor cocktail (78410, Life Technologies). Lysates were sonicated, cleared by centrifugation, and protein concentration determined by BCA protein assay kit (Life Technologies). Samples were denatured by

boiling in loading buffer (7722, Cell Signaling). 30 µg of lysates were loaded on 4–20% Novex trisglycine gels (Life Technologies), transferred to Immobilon (PVDF) membranes (EMD Millipore), blocked in TBST (0.1% Tween-20) and 5% milk. Primary incubation with antibodies was overnight at 4°, followed by appropriate HRP-conjugated secondary (Santa Cruz Biotechnology, Dallas, TX) and detected using ECL plus (32132, Life Technologies) or Femto chemiluminescent substrate (34096, Life Technologies). Antibodies used were obtained from Cell Signaling and raised against the following proteins: phospho p-38 (4511), p38 (8690), p-p44/p42 (4370), p44/p42 MAPK (4695), p-SAPK/JNK (4668), SAPK/JNK (9258), p-EGFR (2234), EGFR (2232), LC3A (4599), p-Akt (13038), Akt (9272), PARP (9542), GAPDH (2118). Additionally, we used antibodies against KRAS (F234, Santa Cruz Biotechnology) and GFP (A-21311, Life Technologies).

## Tests for cell viability

To determine the number of viable cells over a 7-day time course, cells were seeded in triplicate in 6-well plates at 20,000 cells/well (PC9 derivatives) or 40,000 cells/well (H358 derivatives). Cells were seeded into doxycycline (100 ng/ml) and/or erlotinib (1 µM). Media (with or without doxycycline or erlotinib) were not replenished during the 7 days. At indicated time points, media was aspirated and replaced with media containing Alamar Blue (Life Technologies). Fluorescence intensities from each well were read in quadruplicate on a FluoStar Optima instrument (BMG Labtech, Cary, NC) and data plotted in Prism (GraphPad). Cells were counted using Trypan Blue (Lonza) and automated cell counter Countess (Life Technologies).

## Measures of apoptosis

Cells were seeded in the same format as for the 7-day time course. On day 5, media was collected and adherent cells removed by Accutase (eBioscience, San Diego, CA) and cells stained with PE-Annexin V and 7-AAD (BioLegend, San Diego, CA). Cells were analyzed on a BD FACS Calibur and data processed using FlowJo software (FlowJo, Ashland, OR).

## Cell cycle analysis

Cells were seeded in the same format as for cell viability studies. Over a 7-day time course, cell culture media was aspirated and adherent cells removed by Accutase (BioLegend). Cells were washed in PBS and fixed in 66% ethanol on ice, incubated overnight at 4°, pelleted, washed in PBS, and stained in propidium iodide/RNase staining solution (4087, Cell Signaling). Cells were analyzed on a BD FACScan BD Biosciences, San Jose, CA) with a Cytek dxp8 (Cytek, Fremont, CA) upgrade and data processed using FlowJo software.

## Cell imaging

Phase contrast images were taken on a AxioObserver.A1 (Zeiss, Thornwood, NY). For EM, cells were processed and embedded as described (*Gonda et al., 1976*). Cells cultured in a 6-well plate were fixed in 0.1 M cacodylate buffer containing glutaraldehyde (2% vol/vol) for 1 hr at room temperature, washed three times in cacodylate buffer (0.1 M, pH 7.4), and post fixed in osmium tetroxide (1% vol/vol) for 1 hr at room temperature. The cells were stained in 0.5% wt/vol uranyl acetate (0.5% vol/vol) in acetate buffer (0.1 M, pH 4.5) for 1 hr at room temperature. Cells were gradually dehydrated in a series of ethanol solutions (35%, 50%, 75%, 95%, and 100%). Cells were washed in a pure epoxy resin three times after 100% ethanol, and then embedded in the resin. The resin was cured in an oven (55°C) for 48 hr. The cured resin blocks were separated from the plate by submerging in liquid nitrogen. The cells were examined under an inverted microscope and areas selected for preparing 80 to 90 nm sections, which were then mounted on 200 copper mesh grids and counter-stained in uranyl acetate and lead citrate. The grids were carbon coated in a vacuum evaporator. The grids were examined and imaged in the electron microscope operated at 80 kv and digital images were captured by a CCD camera.

## Measurement of macropinocytosis

Cells were seeded in 12-well plate in phenol red-free RPMI (Life Technologies) and 10% Tetracycline-free FBS (Clontech) and doxycycline (100 ng/ml), as indicated. Cells were stimulated with dextran, Oregon Green 488; 70 kDa (D7173, Life Technologies) at 1 mg/ml for 30 min, washed in cold PBS and cells removed by Accutase (eBioscience). Cells were analyzed on a BD FACScan with a Cytek dxp8 upgrade and data were processed using FlowJo software.

## Gene expression profiling

H358 and PC9 cells and their derivatives were grown in 6-well plates, treated with or without 100 ng/ml dox in duplicate for 24 hr, and total RNA was extracted using the RNeasy Mini kit (Qiagen) as described above. Sample quality was assessed using an Agilent Bioanalyzer (Agilent, Santa Clara, CA) and subsequent sample preparation, array hybridization, and data acquisition was performed by the National Human Genome Research Institute's Microarray Core facility. GeneChip Human Gene 2.0 ST microarrays (Affymetrix, Santa Clara, CA) were used according to the manufacture's protocols. Raw data (Affymetrix CEL files) were normalized by robust multiarray analysis (*Irizarry et al., 2003*) and subsequently analyzed to detect genes differentially expressed between dox-treated and untreated cultures, using the ANOVA function in Partek Genomics Suite software (Partek, St. Louis, MO); a Benjamini–Hochberg corrected p value of <0.01 was considered significant. Overlap between differentially expressed genes among the cell lines was determined using VENNY software (*Oliveros, 2007*) and those significantly deregulated upon the addition of dox in both H358-TetO-EGFR$^{L858R}$ and PC9-TetO-KRAS$^{G12V}$ cells and not their respective TetO-GFP controls were selected for further analyses. DNA probes that detected significant differences were mapped to genes using Partek Genomics Suite; those not mapping to annotated genes were removed. An average fold-change for dox treated vs untreated H358-TetO-EGFR$^{L858R}$ and PC9-TetO-KRAS$^{G12V}$ cells was calculated and these values, along with the corresponding probe IDs, were uploaded to IPA software (Ingenuity Systems, Qiagen) and analyzed using the Canonical Pathways function. Canonical pathways with a z-score ≥0.25 and a p value ≤0.05 were considered activated, whereas those with a z-score ≤ −0.25 and a p value ≤0.05 were considered inactivated. GSEA (*Subramanian et al., 2005*) was also performed on the same gene list using the pre-ranked GSEA function and the transcription factor targets gene sets within the C6: Oncogenic Signatures Collection from the Molecular Signature Database (*Subramanian et al., 2005*). Genes were ranked according to their average fold change for the dox treated TetO-Oncogene cells (described above) and GSEA was run with default settings except 'Min size: exclude smaller set = 10'.

## Acknowledgements

We would like to thank Suiyuan Zhang (National Human Genome Research Institute Bioinformatics and Scientific Programming Core) and Raj Chari (Harvard Medical School, Department of Genetics) for their help with the genome-wide data analyses. We would also like to thank Danielle Miller-O'Mard, Jackie Idol and Jeff Chen for technical assistance and Dennis Fei for reading the manuscript.

## Additional information

### Funding

| Funder | Grant reference | Author |
| --- | --- | --- |
| National Institute of General Medical Sciences (NIGMS) | Intramural Program | Arun M Unni, William W Lockwood, Kreshnik Zejnullahu, Shih-Queen Lee-Lin |
| Canadian Institutes of Health Research | | William W Lockwood |

The funders had no role in study design, data collection and interpretation, or the decision to submit the work for publication.

### Author contributions

AMU, WWL, Conception and design, Acquisition of data, Analysis and interpretation of data, Drafting or revising the article; KZ, S-QL-L, Acquisition of data, Analysis and interpretation of data; HV, Conception and design, Analysis and interpretation of data, Drafting or revising the article

### Ethics

Animal experimentation: This study was performed in strict accordance with the recommendations in the Guide for the Care and Use of Laboratory Animals of the National Institutes of Health. All of the animals were handled according to approved institutional animal care and use committee (IACUC) protocols (G-10-6).

## Additional files

### Supplementary files

• Supplementary file 1. KRAS and EGFR Mutations in Lung Adenocarcinoma (Excel File A).

• Supplementary file 2. Combinations of mutated genes significantly associated, positively or negatively, in various cancer types (Excel File B).

• Supplementary file 3. Genotypes for all experimental mice used in this study (Below).

• Supplementary file 4. Genes differentially expressed 24 hr after co-induction of mutant KRAS and EGFR (Excel File C).

• Supplementary file 5. Gene sets enriched upon co-induction of mutant KRAS and EGFR.

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
