## [Decision Letter]

Thank you for sending your work entitled “Synthetic lethality explains mutually exclusive mutations of KRAS and EGFR in lung adenocarcinoma” for consideration at *eLife*. Your article has been favorably evaluated by Charles Sawyers (Senior editor) and two reviewers, one of whom, Chi Dang, is a member of our Board of Reviewing Editors.

The Reviewing editor and the other reviewer discussed their comments before we reached this decision, and the Reviewing editor has assembled the following comments to help you prepare a revised submission.

In this manuscript, Unni et al. explore the potential explanations for an intriguing clinical observation, namely the absence of co-occurring activating mutations in KRAS and EGFR in lung adenocarcinoma despite the relatively frequent occurrence of these mutations. Importantly, they were unable to find similar associations between other commonly occurring mutations in this tumor type suggesting that their co-existence is detrimental. Further the authors demonstrate using transgenic mouse models that simultaneous induction of these mutated oncogenes resulted overwhelmingly with tumors that carry either mutated genes but not both. This exclusivity was further documented in human lung adenocarcinoma cell lines, in which co-expression of these oncogenes resulted in loss of viability, vacuolization, and evidence of macropinocytosis and autophagy. These intriguing results, however, were not further pursued at the mechanistic level, which was explicitly stated to be beyond the scope of the current work. Nonetheless, molecular signatures of cells co-expressing these oncogenes revealed activation of MAPK/JNK pathways. Whether co-expression of these oncogenes resulted in profound metabolic stress would be of great interest in future studies. In particular, whether autophagy is a survival or a death mechanism in this model remains to be established. The reported findings have significance regarding the evolution of human tumors and potential have implications for the application of targeted therapies. The work is high quality and would be of interest to *eLife* readers.

Key issues to address:

1) The authors should provide addition statistical analysis and provide error bars where available. For example, Figure 3 graph for H358KRASG12C-TetOEGFR-L808R shows no error bars (unless they are extremely small).

2) The authors also show other examples of potential “synthetic lethal” interactions including concomitant mutation of BRAF and KRAS in colorectal cancer. In this regard, recent work from Ryan Corcoran and colleagues (Ahronian et al., Cancer Discov. 2015 April;5(4):358-67. doi: 10.1158/2159-8290.CD-14-1518. Epub 2015 Feb 11) indicate that amplification of KRAS is selected for in colorectal cancers treated with a BRAF inhibitor. This suggests that scenarios exist, potentially induced by treatment, where co-activation of otherwise mutually exclusive alterations is selected for as potential resistance mechanisms. The authors' data should therefore be discussed in light of these recent published findings. The implications of these recent findings with regards to therapy should also be discussed.

3) The analysis of gene expression profiles in cancer cells expressing EGFR and KRAS, as well as confirmatory immunoblots, suggest that the activation of MAP kinase signaling is one of the consequences of concomitant oncogene expression. The data suggest that the JNK pathway may be most robustly activated. One possibility therefore is that JNK inhibition would ameliorate the lethality observed following combined EGFR and KRAS activation. This is a straightforward experiment to execute and would provide critical support for the proposed mechanisms underlying the induced lethality. Otherwise, the presented in vitro data are correlative. This issue could also be addressed in the Discussion.

4) The authors suggest that enhanced macropinocytosis could potentially contribute to the reduced fitness following co-expression of EGFR and KRAS. However, given the published data of Commisso et al. (Nature. 2013 May 30;497(7451):633-7. doi: 10.1038/nature12138. Epub 2013 May 12.), which the authors refer to, it is unclear why the authors believe that enhanced macropinocytosis would be detrimental in this case. Further discussion of this point would be beneficial.

---

## [Author Response]

*1) The authors should provide addition statistical analysis and provide error bars where available. For example,*
Figure 3
*graph for H358KRASG12C-TetOEGFR-L808R shows no error bars (unless they are extremely small)*.

We have now added statistical information and corresponding details within the figures and figure legends throughout the manuscript. We should note that for Figure 3, there are error bars for H358KRASG12C-TetOEGFR-L858R but they are small. The data trend—loss of viability with time—is representative of independent experiments.

*2) The authors also show other examples of potential “synthetic lethal” interactions including concomitant mutation of BRAF and KRAS in colorectal cancer. In this regard, recent work from Ryan Corcoran and colleagues (Ahronian et al., Cancer Discov. 2015 April;5(4):358-67.* doi: 10.1158/2159-8290.CD-14-1518*. Epub 2015 Feb 11) indicate that amplification of KRAS is selected for in colorectal cancers treated with a BRAF inhibitor. This suggests that scenarios exist, potentially induced by treatment, where co-activation of otherwise mutually exclusive alterations is selected for as potential resistance mechanisms. The authors' data should therefore be discussed in light of these recent published findings. The implications of these recent findings with regards to therapy should also be discussed*.

The reviewers bring up an important point regarding the mutual exclusivity of other oncogenes and the influence of treatment on these observed relationships. First, it should be noted that our study looked only at sequence level mutation and not copy number status, an important consideration in the context of the work by Corcoran and colleagues. Thus, we would not record the co-incidence of oncogenic mutation of BRAF with amplification of KRAS in colorectal cancers or similar relationships between oncogenes in other cancer types. However, it should be noted that mutation and amplification, while both “activating” in their effect on oncogenes, work through different mechanisms and are likely to have different effects on signaling. For example, unlike lung adenocarcinomas with sequence-altering mutations of EGFR, tumors with amplifications of EGFR do not acutely respond to EGFR tyrosine kinase inhibitors. This finding and others suggest that mutations in EGFR potentiate signaling to a greater degree than amplification of wild-type EGFR. Therefore, in the context of synthetic lethality, when signaling intensity may need to exceed thresholds to induce toxicity, a combination of mutation of one oncogene and amplification of another may not create mutual exclusivity even though other mutational combinations might do so. Thus, as described in the paper by Corcoran and colleagues, a pre-existing population of cells within a tumor may carry both a BRAF mutation and KRAS amplification and those cells would have a selective advantage that upon treatment with BRAF inhibitors. Likewise, since our study examined tumors before treatment, it is possible that oncogenic alleles that appear to be excluded by the presence of certain mutations in the cancer types we examined may occur and be tolerated upon treatment with targeted therapies. Since we tested directly only the lethality of EGFR and KRAS mutations, we cannot exclude these and other possibilities. We pointed out in our original manuscript the need for such direct tests of the mechanism of observed mutual exclusivities.

To clarify the scope of our analysis, the important distinctions between amplifications and mutations, and the influence of factors such as drug treatment and resistance, we have added a paragraph to the Discussion and included a reference to Corcoran et al.

*3) The analysis of gene expression profiles in cancer cells expressing EGFR and KRAS, as well as confirmatory immunoblots, suggest that the activation of MAP kinase signaling is one of the consequences of concomitant oncogene expression. The data suggest that the JNK pathway may be most robustly activated. One possibility therefore is that JNK inhibition would ameliorate the lethality observed following combined EGFR and KRAS activation. This is a straightforward experiment to execute and would provide critical support for the proposed mechanisms underlying the induced lethality. Otherwise, the presented in vitro data are correlative. This issue could also be addressed in the Discussion*.

We agree with the reviewers’ suggestion that additional experiments are required to determine whether the enhanced kinase activities of JNK and p38 cause synthetic lethality or are simply correlated with the biological effects, even though a link between these pathways and the phenotype observed after co-expression of KRAS and EGFR (increased macropinocytosis and vacuolization) is reported in the literature. To determine if inhibition of JNK and p38 can rescue the lethality observed upon co-expression of mutant EGFR and KRAS, we performed several experiments with siRNAs and small molecule inhibitors directed at these kinases in cells, using the inducible cell lines previously deployed in Figure 5 of our original manuscript. While we observed some amelioration of the lethal effects of induction of a second oncogene, there were confounding issues with these experiments that precluded adding them to the manuscript. First, while described as specific inhibitors for JNK and p38, the compounds used, like other small molecules, are known to affect the activity of other proteins (kinases) as well. Thus, it is difficult to attribute the effects observed solely to inhibition of the targeted kinases and not additional off-target effects. Second, as both JNK and p38 have multiple family members (i.e. JNK1-3 and p38 alpha, beta, gamma and delta) that mediate their signaling cascades, we could not reliably silence all family members concurrently with siRNAs.

Addressing these two experimental issues, by optimizing conditions, testing more compounds, and monitoring effects on multiple kinases, would require substantial efforts and unnecessarily further delay the publication of our central findings regarding mutual exclusivity and synthetic lethality. Therefore, while it is important to establish the downstream events mediating the observed lethality, these experiments are better suited to follow-up studies.

However, to add caveats about the association of JNK and p38 hyperactivity with synthetic lethality, we have added text to the Discussion to describe the correlative nature of the observations and the need for additional experiments.

*4) The authors suggest that enhanced macropinocytosis could potentially contribute to the reduced fitness following co-expression of EGFR and KRAS. However, given the published data of Commisso et al. (Nature. 2013 May 30;497(7451):633-7.* doi: 10.1038/nature12138*. Epub 2013 May 12.), which the authors refer to, it is unclear why the authors believe that enhanced macropinocytosis would be detrimental in this case. Further discussion of this point would be beneficial*.

We have clarified what we believe to be the difference between “controlled” macropinocytosis such as that observed by Commisso et al. and the “uncontrolled” macropinocytosis observed in cells expressing mutant KRAS and mutant EGFR. To summarize, we believe that the increased signaling induced by co-expression of both mutant oncogenes promotes macropinocytosis to levels or durations greater than those seen with expression of KRAS alone, resulting in profound cell defects such as vacuolization (Discussion).